# COVID-19-Related Changes in Perceived Household Food Waste in the United States: A Cross-Sectional Descriptive Study

**DOI:** 10.3390/ijerph18031104

**Published:** 2021-01-27

**Authors:** Kelly Cosgrove, Maricarmen Vizcaino, Christopher Wharton

**Affiliations:** Radical Simplicity Lab, College of Health Solutions, Arizona State University, 550 N 3rd Street, Phoenix, AZ 85004, USA; mvizcain@asu.edu (M.V.); Christopher.Wharton@asu.edu (C.W.)

**Keywords:** food waste, COVID-19, household, sustainability, United States

## Abstract

Food waste contributes to adverse environmental and economic outcomes, and substantial food waste occurs at the household level in the US. This study explored perceived household food waste changes during the COVID-19 pandemic and related factors. A total of 946 survey responses from primary household food purchasers were analyzed. Demographic, COVID-19-related household change, and household food waste data were collected in October 2020. Wilcoxon signed-rank was used to assess differences in perceived food waste. A hierarchical binomial logistic regression analysis was conducted to examine whether COVID-19-related lifestyle disruptions and food-related behavior changes increased the likelihood of household food waste. A binomial logistic regression was conducted to explore the contribution of different food groups to the likelihood of increased food waste. Perceived food waste, assessed as the estimated percent of food wasted, decreased significantly during the pandemic (*z* = −7.47, *p* < 0.001). Food stockpiling was identified as a predictor of increased overall food waste during the pandemic, and wasting fresh vegetables and frozen foods increased the odds of increased food waste. The results indicate the need to provide education and resources related to food stockpiling and the management of specific food groups during periods of disruption to reduce food waste.

## 1. Introduction

Coronavirus disease 2019 (COVID-19) was declared a global pandemic on 11 March 2020 [1]. On 16 March 2020, a nationwide “stay-at-home” order was issued in the United States, marking the beginning of a period of dramatic disruptions in the daily lives of most people [2]. Although such change is likely to have impacts on a variety of behaviors that contribute to the well-being of households and their members, existing studies on the impacts of COVID-19 lockdowns on various food- and health-related behaviors have shown mixed results; some studies have shown improvements in these behaviors [3,4], while others have revealed the adoption of less desirable lifestyle habits [5,6,7].

Household management of food to an extent underlies the ability of members of households to engage in healthful, efficient, or even more or less sustainable food-related behaviors, especially as it relates to food procurement and food use [8,9]. However, the ways in which households purchase and store food, as well as make use of it over time, are subject to COVID-related impacts, for instance in driving individuals to potentially “stockpile” food or alter home-cooking patterns [10,11,12]. As such, it is possible that pandemic-related shifts could also have occurred in relation to household-level food waste. This is a potentially important area to explore given that food waste is increasingly recognized as contributing to adverse environmental and economic outcomes, so much so that the United Nations identified a 50% reduction in global food waste as one of the 17 Sustainable Development Goals in its 2030 agenda [13]. In the US, specifically, it has been estimated that the environmental, social, and economic costs of food waste result in losses amounting to US $2.6 trillion annually [14]. The aggregate of wasted food in the US also represents considerable misuse of resources, with the production of wasted food estimated to require 30 million acres of cropland, substantial amounts of fresh water, and other agricultural inputs such as pesticides and fertilizers [15]. At the consumer and retail levels, roughly 95% of food that is discarded ends up in landfills, where its breakdown results in the release of tens of millions of tons of greenhouse gases, including methane [16].

A substantial portion of food waste occurs at the household level, especially in developed countries, indicating an important target for food-waste-reduction interventions [17]. As a result, literature in this area has grown in recent years, and a recent National Academies report summarized current household food waste research alongside targets for future work [18]. However, the impact of COVID-19 on household- and consumer-level food waste represents a key area of exploration, not only because of the potentially acute impact of the COVID-19 global pandemic, but also in relation to how food- and food-waste-related behaviors may change given any such global disruption in the future.

A small number of studies in English focusing on food waste during the COVID-19 pandemic already have been published. One study conducted in Italy during the nationwide lockdown there indicated that the majority of households reported reduced food waste during the initial lockdown period [19]. Another Italian study reported similar results and indicated that the reduction in food waste was largely attributed to concern related to the waste management system [20]. Other research conducted in Tunisia reported an overall reduction in food waste at the household level during the first two weeks of lockdown [21]. Although most of the studies conducted on household food waste during the COVID-19 pandemic have relied on self-reported data, in a recent study conducted in Malaysia, household waste that was weighed by the waste management department was used as an objective proxy of food waste. This study estimated that there was a 15.1% decrease in food waste during the lockdown period in Malaysia [22].

However, other authors have indicated that the COVID-19 pandemic could eventually lead to increased food waste as a result of panic buying and a lack of proper food planning [23]. Additionally, concerns have been raised regarding the management of overall waste during the COVID-19 pandemic; for example, one study found that overall household waste increased during the COVID-19 pandemic in countries implementing lockdown measures [24]. The authors proposed that this increase was related to the increased use of single-use products and panic buying.

In the US, a number of household factors potentially associated with food-related behaviors were likely to have changed dramatically during the COVID-19 pandemic. However, to the authors’ knowledge, no study has assessed COVID-19-related changes in food waste in the US. Therefore, the primary aims of the study were (1) to examine if perceived household food waste changed during the COVID-19 pandemic and (2) to explore whether lifestyle disruptions and changes in food-related behaviors influenced household food waste during the COVID-19 pandemic among a sample of US adults. In addition, we explored the contribution of different food groups to household food waste changes during the COVID-19 pandemic.

## 2. Materials and Methods

### 2.1. COVID-19 and Food Waste Questionnaire

A 57-item questionnaire was used to collect information on individual demographic factors of the household food purchaser (e.g., age, sex, education), household characteristics (e.g., income, household size, number of children under the age of 18 years), COVID-19-related household changes (e.g., children staying at home because of the COVID-19 pandemic, changes in employment income as a result of COVID-19), and changes in food-related behaviors due to the pandemic (e.g., food stockpiling, frequency of home food preparation). Demographic items reflected those used in the US Census Pulse Surveys to gather similar data [25].

For household food waste estimation, participants were asked to estimate the percentage of food purchased at the grocery store that they threw away in an average week using a sliding scale from 0% to 100%, which has been shown to be a valid method to estimate household food waste in prior research [26]. Participants estimated their household food waste both before the COVID-19 pandemic and during the COVID-19 pandemic. Additionally, participants were asked about the specific waste of different food groups during the pandemic. Participants were presented with the question “Compared to your habits prior to the COVID-19 pandemic, how have your food waste behaviors changed during the pandemic?” A comprehensive list of 12 different food groups was provided, including fresh fruits, fresh vegetables, dairy products, meat/poultry, among others. Participants were asked to respond on a Likert-type scale ranging from 1 = waste significantly less to 5 = waste significantly more.

### 2.2. Data Collection

Participants were recruited via Amazon Mechanical Turk (MTurk), and the survey was delivered online through Qualtrics. Previous studies have indicated that individuals residing in the US who complete surveys on MTurk are more similar to the US internet population and more diverse than traditional pools of subjects (e.g., college students) [27]. Additionally, surveys conducted with participants recruited via MTurk have been shown to provide valid and reliable data in the context of health and social sciences [28,29]. Participants qualified for the study if they were adults aged 18 years or older who currently resided in the United States, if they were fluent in the English language, and if they identified as the primary household food purchaser.

Survey responses were collected on 2 October 2020, approximately seven months after US lockdowns began, allowing for the assessment of longer-term lifestyle changes associated with the pandemic. Strategies to ensure the collection of high-quality data were implemented (e.g., attention checks and spam/automated response checks). The study was approved by the Institutional Review Board of Arizona State University (protocol code STUDY00012660), and all participants provided informed consent prior to the completion of the survey.

### 2.3. Statistical Analyses

Data are presented as frequencies and percentages for nominal variables and as medians and interquartile ranges (IQRs) for continuous variables. Data were weighted by age and sex using information provided by the US Census Bureau to make our results representative of American households. All continuous variables were non-normally distributed, even after transformation attempts; therefore, nonparametric analyses were used for inferential tests. The Wilcoxon signed-rank test was used to assess differences in perceived food waste before and during the pandemic. A hierarchical binomial logistic regression analysis was conducted to examine if lifestyle disruptions and changes in food-related behaviors due to the COVID-19 pandemic increased the likelihood of household food waste. Covariates included age, sex, education, income, total number of members in the household, and presence of children in the household. Lifestyle changes included income loss, childcare changes, and adoption of telework. Food-related behaviors included percentage of meals prepared at home and food stockpiling. Lastly, a binomial logistic regression was conducted to explore the contribution of different food groups to the likelihood of increased food waste. All statistical analyses were conducted using SPSS v. 27 (IBM Corp., Armonk, NY, USA). Statistical significance was set at alpha level <0.05.

## 3. Results

### 3.1. Demographics and Household Characteristics

A total of 946 individuals residing in the United States who self-identified as the primary food purchaser in their household completed the survey. The majority of respondents were male (51.4%), had a bachelor’s degree (59.4%), and were married (65.9%) (see Table 1). Households with an income between $50,000 and $74,999 (28.9%), four household members (29.0%), and no children (37.7%) were predominant (see Table 2).

### 3.2. COVID-19-Related Household Changes

The COVID-19 pandemic has resulted in substantial changes in childcare circumstances and employment and working conditions, with a considerable shift toward telework (Figure 1) [30]. In this study, 45.6% of household food purchasers reported that at least one adult in their household substituted *all* of their in-person work with telework, 26.9% reported that at least one adult substituted *some* of their in-person work with telework, and 27.5% reported that no adults in their household substituted their in-person work with telework. Despite these shifts toward telework, 48.2% of households included in the study reported that at least one household member experienced COVID-related income loss. Further, in 87.0% of households with children (*n* = 545), an adult in the household was reported to be responsible for caring for children who were staying at home because of COVID-19.

### 3.3. COVID-19-Related Food Behavior Changes

Survey respondents reported consuming a greater percentage of home-prepared meals during the pandemic than before the pandemic. Prior to the pandemic, the median percent of meals prepared at home was 75% (60–90%), whereas this figure increased to 86% (70–95%) during the pandemic (see Figure 2). Furthermore, 73.0% of participants reported stocking up on nonperishable food in response to the COVID-19 pandemic. Of the participants who reported stockpiling nonperishable foods, 23.1% purchased much more food than normal, 51.0% purchased moderately more food than normal, and 25.9% reported buying slightly more than normal (see Figure 3). Approximately one-third of participants who reported stockpiling food expected that all of the food they purchased when stocking up would eventually be consumed by people in their household (see Figure 4).

### 3.4. COVID-19-Related Household Food Waste Changes

Overall, 50.8% of participants reported decreased food waste, 26.5% reported increased food waste, and 22.7% reported no change in food waste during the COVID-19 pandemic. Results from the Wilcoxon signed-rank test indicate that the median percent of household food waste significantly decreased from 20% to 10% during the COVID-19 pandemic compared to before the pandemic across the full sample (*z* = −7.93, *p* < 0.001). Additional subgroup analyses indicate that changes in household food waste varied by age group. Adults between the ages of 20 and 29 years reported an increase in perceived food waste; however, the increase was not statistically significant (*Mdn_before_* = 35.26% vs. *Mdn_during_* = 38.0%, *z* = −1.66, *p* = 0.09). In contrast, adults in all the other categories reported a decrease in perceived food waste; these changes were all statistically significant (all *p* < 0.001). Because some participants reported increased food waste during the pandemic and in alignment with the second aim of our study, we explored the factors associated with increased food waste, as households with characteristics associated with increased food waste might benefit from specifically targeted food waste reduction interventions.

### 3.5. Impact of Lifestyle Disruptions and Food-Related Behaviors on Household Food Waste

Based on the results above, the hierarchical binomial logistic regression was conducted by excluding participants that reported no change in perceived food waste. Subsequently, the dichotomy of increase/decrease was used as the dependent variable. The model was significant (χ^2^(11) = 54.16, *p* < 0.001). Covariates alone explained 7.6% of the variance in food waste; the only significant predictors included age and sex, which indicate that being older and a female decreased the odds of greater food waste by factors of 0.97 and 0.49, respectively. The remaining covariates (e.g., income and education) were not significant (all *p* > 0.26). Adding the variables associated with lifestyle disruptions and changes in food-related behaviors due to the COVID-19 pandemic into the model increased the proportion of the food waste variance explained to 12.6%. However, the only significant predictor was food stockpiling, which increased the odds of greater food waste by a factor of 1.7.

### 3.6. Contribution of Different Food Groups to Household Food Waste

The binomial logistic regression model examining the contribution of the different food groups to the likelihood of greater food waste was significant (χ^2^(12) = 117.07, *p* < 0.001). The model explained 20.3% of the variance in food waste. Food groups contributing significantly to the model included fresh vegetables and frozen foods, which increased the odds of greater food waste by factors of 1.29 and 1.30, respectively.

## 4. Discussion

The COVID-19 pandemic has altered the daily lives of people worldwide. Although these changes could be viewed as negative in many ways for different groups of people, the disruption also has presented a unique opportunity to understand how altered household circumstances might affect the adoption of more or less healthful and sustainable behaviors. Previous studies have indicated that certain lifestyle-related outcomes, such as diet quality and food waste, have improved during the COVID-19 pandemic [3,4]. However, other recent studies revealed that the pandemic resulted in adverse effects on lifestyle behaviors [5,6,7]. The pandemic therefore represents a vital and timely opportunity to explore the differences between individuals whose lifestyles changed in potentially favorable ways and those whose lifestyles were adversely impacted by the pandemic.

The results of the present study are in accordance with those of previous studies examining COVID-19-related food waste in other countries [19,21]. In a study conducted in Qatar, researchers found that self-reported household food waste decreased overall. However, in the Qatari study, 44.81% of respondents reported wasting less food, and 41.61% reported that their self-reported food waste did not change, indicating that food waste increased in only 13.58% of households in Qatar during the COVID-19 lockdown [31]. However, in the present study, 28.2% of households reported increased food waste during the COVID-19 pandemic. This difference could be related to the timing of survey conduction: the present study was conducted in October 2020, seven months after the start of the pandemic, whereas the Qatari study was completed in June 2020, only three months after the start of the pandemic. Potential longitudinal studies on COVID-19-related food waste should be conducted to elucidate how food waste trends have changed throughout the COVID-19 pandemic. Another potential explanation for this divergence of findings of the prevalence of increased food waste may be cultural differences. A recent study conducted in Japan, for example, indicated that cultural differences play an important role in food waste behaviors [32].

In the present study, food stockpiling was identified as a significant predictor of increased food waste during the COVID-19 pandemic even after controlling for household and food purchaser characteristics. Food stockpiling during the COVID-19 pandemic has been shown to be associated with the need for a sense of stability and predictability during uncertain times, leading to the adoption of an individualistic strategy associated with buying more even if it leads to food waste or reduced availability of food for others [33]. This finding has important implications regarding food policy, crisis communication, and food purchasing habits, providing an important target for educational interventions with the aim of decreasing food waste and increasing food security during crisis situations. In preparation for the COVID-19-related lockdown in the United States, people were encouraged to stock up on food to reduce the likelihood of having to leave home during the lockdown period. While these recommendations were necessary, in the future, providing consumers with effective guidelines to follow when stocking up on food could reduce the amount of food that would be wasted as a result of panic buying. For example, providing guidelines regarding how much food to purchase, what types of food to purchase, and how to use the purchased food would be useful to guide food acquisition.

The findings of this study showed that over 75% of participants purchased moderately more to much more food than normal due to COVID-19 and that only one-third of participants thought that all of the food purchased when stocking up would eventually be consumed by people in their household. This result has substantial food waste implications that warrant further investigation. Specifically, it is possible that although food waste seems to have decreased during the COVID-19 pandemic, there is still potential for food waste to increase in the longer term as a result of stockpiled food going to waste. Additionally, the result indicates that people either did not stock up on appropriate foods or did not have the means or desire to consume all the food they purchased when stocking up. These are two important targets for educational campaigns and food policy in future crisis situations as well as during the ongoing COVID-19 pandemic. For example, in future situations in which it is recommended that people stock up on food, such as prior to shelter-in-place orders or expected major weather events, providing additional resources to guide people’s shopping and consumption habits could lead to more strategic grocery purchases that contribute to more efficient utilization, and thus meaningful reductions in food waste associated with stockpiling. Additionally, recommendations for how to utilize commonly stockpiled items could lead to meaningful food waste reduction in crisis situations. Recommendations for food stockpiling in emergency situations have been developed. For example, in 2007, *The Medical Journal of Australia* published important information on what types of food to include in a disaster-preparedness stockpile in regard to nutrient and energy requirements, quantity, space, shelf stability, etc.; however, the article did not mention the importance of rotating stockpiled goods to ensure that food is consumed before it expires [34].

The present results related to stockpiling were different from those of prior studies. In Qatar, researchers reported that 73.34% of respondents reported not stocking up on food during the COVID-19 pandemic, whereas only 26.5% of respondent in the present study reported not stocking up on food during the pandemic. Interestingly, the Qatari study revealed that there was a significant difference in the prevalence of food stockpiling according to citizenship status: 78.2% of Qatari respondents and 65.09% of non-Qatari respondents reported not stocking up on food [31]. This finding suggests that there may be a cultural difference in food stockpiling behaviors. For example, previous studies have identified differences in crisis response between individualistic and collectivistic societies; individuals in individualistic societies have been shown to be more likely to adopt behaviors associated with individual gains and adverse collective outcomes, which would include stockpiling of resources [35]. This might explain the high prevalence of food stockpiling observed in the present study conducted in the US, an individualistic society.

The results of the present study also identified food purchaser sex as a predictor of changes in perceived food waste. Previous studies have shown similar trends in food waste according to food purchaser sex. For example, in a study conducted in Denmark, it was found that male sex was associated with more food waste [36]. A study that explored consumer food waste among Romanians according to gender revealed that women tended to be more concerned about the negative impacts of food waste than men [37]. This sex difference in the attitude toward food waste could explain the differences in food waste between men and women reported in the current study. Moreover, differences in food waste according to age group were identified. Previous studies have also indicated that food waste differs according to age [38,39]. There are many possible explanations for this phenomenon. For example, younger people likely have less experience with food procurement and management, which could lead to higher waste. Additionally, older individuals may see food as more valuable if they have gone through economically challenging periods in which food was not as easily accessible, or they may have been raised to consider food to be more valuable.

An additional finding of this study that has important food waste implications is that fresh vegetables and frozen food were found to contribute significantly to increased food waste during the COVID-19 pandemic. This finding indicates the potential need to develop interventions to encourage individuals to improve their management of these foods by buying only what will be consumed, planning meals in advance, and understanding how to properly store and use leftovers. Additionally, it is common for fresh vegetables to be reported as one of the main food waste categories due to their short shelf-life. However, the finding that frozen food contributed significantly to increased food waste during the COVID-19 pandemic deserves further attention to examine what led individuals to discard frozen foods. Perhaps additional exploration in this area could allow for the dissemination of useful information related to frozen food purchasing and storing that could be easily implemented at the household level, leading to meaningful reductions in food waste in this category.

This study benefitted from a number of strengths but also suffered from some limitations. The study included a relatively large national sample. Additionally, to the authors’ knowledge, this was the first study to explore the impact of the COVID-19 pandemic on household food waste in the US. As a developed country, the US is known for its high level of food waste, which presents an important opportunity for meaningful change. Even so, this was a cross-sectional study, and as such, causal relationships among the factors analyzed cannot be determined. Additionally, food waste was self-reported by participants, and the amounts reported could be influenced by social desirability bias or a lack of understanding of what is considered food waste. However, a measure for perceived household food waste that has been previously found to significantly correlate with actual collected and weighed food waste among US households was used [26].

Participants were also asked to report how much food was wasted in their household prior to the start of the pandemic, which could lead to inaccurate estimations due to recall bias. Despite these limitations, this study provided meaningful insight into the household characteristics that are associated with COVID-19-related changes in perceived food waste. Future studies should identify specific food-waste-related targets to design effective interventions related to meal planning, food storage, food buying, and management of leftovers that could potentially be targeted specifically among households with certain characteristics to further increase their effectiveness.

This study provided important information regarding changes in perceived food waste during the COVID-19 pandemic, and future research should examine additional factors that could contribute to changes in food-related behaviors, such as children in the household transitioning to distance learning and individuals in the household receiving federal assistance, as these factors can impact food acquisition and thus food waste.

## 5. Conclusions

This exploratory study revealed that perceived food waste has decreased overall during the COVID-19 pandemic in the US. This study provided important findings to guide further studies related to food waste changes and stockpiling habits in the US. These findings are important for future societal disruptions as well as the ongoing pandemic, which could lead to long-term lifestyle changes, such as an increase in telecommuting, changes in food purchasing habits, and differences in home food preparation habits.

## Figures and Tables

**Figure 1 ijerph-18-01104-f001:**
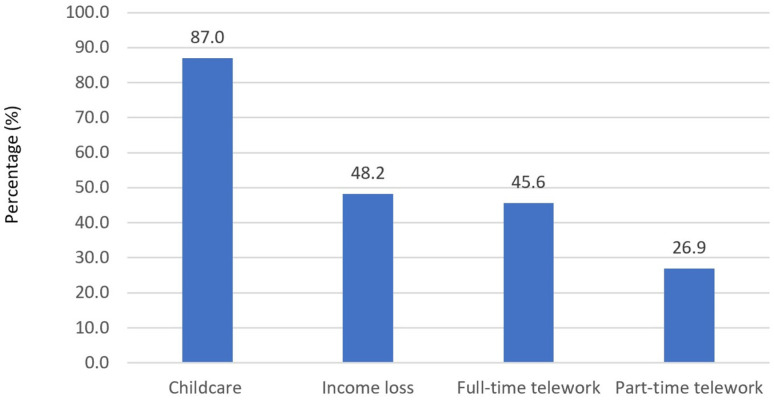
Percentage of participants reporting COVID-19-related household changes. Note. Childcare represents adults in the households with children responsible for caring for children staying at home because of the COVID-19 pandemic. Income loss represents adults experiencing a loss of employment income due to the COVID-19 pandemic. Telework represents adults experiencing changes in typical in-person work because of the COVID-19 pandemic.

**Figure 2 ijerph-18-01104-f002:**
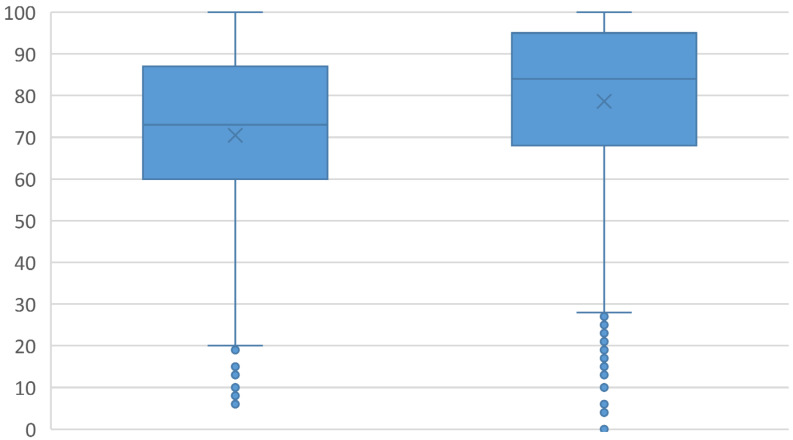
Changes in the percentage of meals prepared in the household before and during the COVID-19 pandemic.

**Figure 3 ijerph-18-01104-f003:**
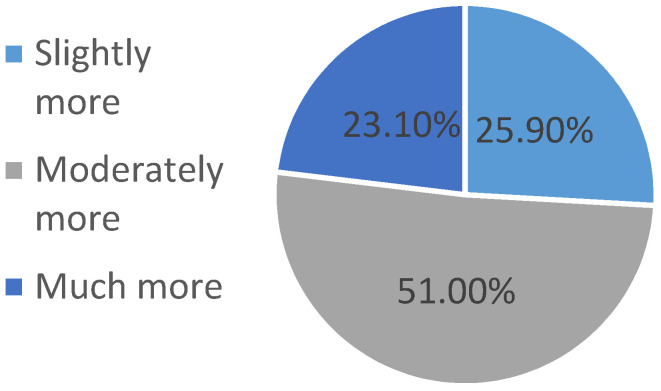
Amount of extra food purchased among participants who reported stockpiling.

**Figure 4 ijerph-18-01104-f004:**
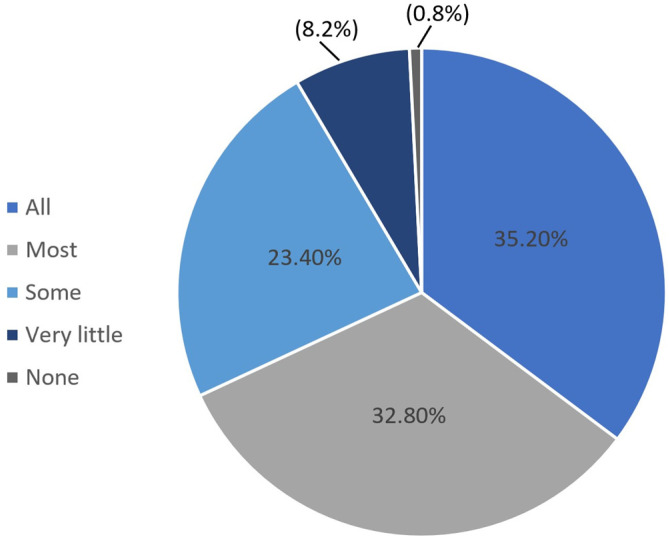
Reported amount of stockpiled food that will eventually be consumed by someone in the household.

**Table 1 ijerph-18-01104-t001:** Demographic characteristics of US adults who completed the COVID-19 and Food Waste Questionnaire (*N* = 946).

	Frequency	Percentage
Sex		
Male	486	51.4
Female	460	48.6
Age range (years)		
20–29	199	21.0
30–39	331	35.0
40–49	200	21.1
50–69	216	22.8
Education		
Less than high school	3	0.3
Some high school	1	0.1
High school graduate	48	5.1
Some college	103	10.9
Associate’s degree	83	8.8
Bachelor’s degree	562	59.4
Graduate degree	146	15.4
Marital status		
Married	623	65.9
Widowed	23	2.4
Divorced	63	6.7
Separated	15	1.6
Never married	222	23.5

**Table 2 ijerph-18-01104-t002:** Household characteristics of US adults who completed the COVID-19 and Food Waste Questionnaire (*N* = 946).

	Frequency	Percentage
Household income		
Less than $25,000	124	13.1
$25,000–$34,999	89	9.4
$35,000–$49,999	172	18.2
$50,000–$74,999	273	28.9
$75,000–$99,999	153	16.2
$100,000–$149,999	96	10.1
$150,000–$199,999	22	2.3
$200,000+	17	1.8
Household size		
1	129	13.7
2	215	22.8
3	183	19.4
4	274	29.0
5+	143	15.1
Number of children		
0	357	37.7
1	263	27.8
2	246	26.0
3	41	4.3
4+	39	4.1

## Data Availability

The data presented in this study are available from the corresponding author upon reasonable request.

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
