# Peer review of "COVID-19-Related Changes in Perceived Household Food Waste in the United States: A Cross-Sectional Descriptive Study"

_ijerph, 2021, doi:10.3390/ijerph18031104_

Round 1

Reviewer 1 Report

The paper deals with very important issue - food waste during covid 19 pandemic. This paper  analyses perceived household food waste changes during the COVID-19 pandemic and related factors. A total of 958 survey responses from primary household food purchasers were analyzed. The paper provides interesting results however policy implications of conducted study are missing. The introduction needs more references to provide background of the study. In the end of manuscript the structure of manuscript should be provided. The paper lack literature review section therefore, I would advice authors to follow IMRAD profile.

Reviewer 2 Report

The topic of waste production concerns not only the time of pandemic. Making people aware of how food habits have changed with the arrival of this specific period is an interesting approach to the topic. In further scientific work it would be worthwhile to take up the topic in terms not only of the USA but also of other countries from other continents.

Reviewer 3 Report

This manuscript describes COVID-19-related changes in perceived household food waste in the United States by using a cross-sectional descriptive study. The research question is unique and relevant due to the current global pandemic that is occurring. There is a solid rationale for this study based on the reports of substantial food waste in developing countries such as the United States. The authors provide a compelling need for the study based on their report that no study has assessed COVID-19-related changes in food waste in the United States.

The method section is sound. It can be enhanced by considering other factors associated with COVID-19-related food waste in the United States. The authors explore a pertinent list of participant characteristics that could be associated with food waste. In addition to telework as a factor, another important consideration is the shift to distance learning, which may affect access to food among children in the household. For instance, meals that would have been received in school may be sent home. Another important consideration may be receipt of federal assistance which may impact purchase of food. Therefore, the authors can consider these additional factors.

Overall, this is an interesting, novel, and insightful study. The study is unique because the authors report that while there are a small number of studies focusing on food waste during the COVID-19 pandemic in areas around the world, there are no studies that have assessed COVID-19-related changes in food waste in the United States. Therefore, attending to clarifying questions, including about the intervention design, may help to improve the paper.

lines 130-132- These are a pertinent list of covariates.  In addition to telework as factor, another important consideration is shift to distance learning, which may affect access to food among children in the household.  For instance, meals that would have been received in school may be sent home.

Lines 130-132- Another important consideration may be receipt of federal assistance which may impact purchase of food.

Line 143-144 - As the authors mention, a different income bracket may produce different study results.

Overall, this is an interesting, insightful, and relevant study.

Reviewer 4 Report

Materials and methods.

Please provide the questionary.

Lines 96-97. “Participants estimated their household food waste both before the COVID-19 pandemic and during the COVID-19 pandemic”. I’m very sceptical about the experimental approach because respondents were asked to recall something that happened at least 7 months ago (line 115).

Line 119. Please name the mentioned “major southwestern university” that “approved” the current study.

Results

Line 141. Please present numerical data instead of “majority”.

Lines 145-146. “this sample cannot be assumed to be representative of the general population in the United States.”. This goes against the title and objectives.

Discussion

I could not evaluate this section as it is not clear what population category it should represent.

Conclusions

Lines 324-325. “This exploratory study revealed that perceived food waste has decreased overall during the COVID-19 pandemic in the US”. This again goes against the disclosure “this sample cannot be assumed to be representative of the general population in the United States.” (lines 145-146).

Reviewer 5 Report

The article presents an interesting and current problem of the impact of the SARS-COV 2 pandemic on social behavior, in this case in the context of food waste. There are more and more similar studies and they come from different countries or regions in the world, many of them are written in their native language, hence the authors' statement (line 66) that 'a small number of studies focusing on food waste during the COVID-19 pandemic 66 already have been conducted'. However, it would be better to write 'a small number of studies in English ... have been published already'. Nevertheless, the topic  fills the gap in this respect and deserves dissemination. However, it would be good to make some minor adjustments before the final version of the article appears. The main comments are presented below.

  1. It is good that when collecting data and presenting research results, the authors compared two periods - before and after the pandemic started. Nonetheless, it would be worthwhile to make a deeper assessment of the changes during the pandemic. Were changes in consumer attitudes noticed in the initial period of the lockdown and after several months of such a situation? Many studies indicate different behavior of citizens in the first and second phase of the pandemic, for example in the context of making food stockpiling. Was this also observed in the case of the United States?
  2. It is worth supplementing Table 1 and Table 2 with the percentage data  of the total population in the United States. Such information should be available within the US census. Thanks to this, the structure of the observed group could be compared with the average results for the population. If they were similar, the conclusions from the study would have a higher cognitive value.
  3. It is indicated that women declared less food waste during pandemic, as opposed to men. Similarly, there were distinctions for different age groups. How do the authors explain this phenomenon? Please formulate your own hypothesis.
  4. Section 3.4: why there is no reference to other demographic characteristics (only gender and age). After all, the authors also measured other variables such as education, income, household size, and number of children. Even if in other cases there were no confirmed and significant relationships, it is worth mentioning.
  5. The inference should be improved. On the one hand, the authors indicate that in the pandemic conditions, food waste decreased, and on the other, that food stockpiling was identified as the only significant variable determining food-related behaviors. These two conclusions are inconsistent with each other, because food stockpiling should increase waste, which contradicts the results. Please explain it and make it clear.

Detailed comments:

  • I suggest adding 'United States' to key words to define spatial scope of the study;
  • there is no need to repeat the same information in the Introduction and Discussion sections. I am referring to research from Italy and Tunisia (references 19 and 20);
  • I propose to transfer the paragraph with 'limitations' (line 302 and next) from Discussion to Conclusion section (this form can be found most often).

Round 2

Reviewer 1 Report

The paper was significantly improved.

Author Response

Thank you very much for your positive comment as well as your previous suggestions that helped us improve our paper.

Reviewer 4 Report

Through the text, please try to evade splitting words between rows by leaving only 2 letters on the previous row, as lines 59-60 “na-tional”, lines 62-62 “po-tentionally”, etc.

Line 352. Please, move Limitations to Discussion.

Author Response

Thank you for your comments. We have moved the limitations section to the discussion and ensured that no words were split leaving only two letters on the previous line. Thank you very much for this suggestion; it very much improved the readability of the paper.